# On the increased climate sensitivity in the EC-Earth model from CMIP5 to CMIP6

Klaus Wyser[1], Twan van Noije[2], Shuting Yang[3], Jost von Hardenberg[4,5], Declan O'Donnell[6], Ralf Döscher[1]

[1]Rossby Centre, Swedish Meteorological and Hydrological Institute (SMHI), 601 76 Norrköping, Sweden
[2]Royal Netherlands Meteorological Institute, Netherlands
[3]Danish Meteorological Institute (DMI), 2100 Copenhagen, Denmark
[4]Department of Environment, Land and Infrastructure Engineering, Politecnico di Torino, Turin, Italy
[5]Institute of Atmospheric Sciences and Climate, Consiglio Nazionale delle Ricerche (ISAC-CNR), Torino, Italy
[6]Finnish Meteorological Institute, Helsinki, Finland

*Correspondence to*: Klaus Wyser (klaus.wyser@smhi.se)

**Abstract.** Many modelling groups that contribute to CMIP6 (Coupled Model Intercomparison Project phase 6) have found a larger equilibrium climate sensitivity (ECS) with their latest model versions compared to the values obtained with earlier versions for CMIP5. This is also the case for the EC-Earth model, and in this study we investigate what developments since the CMIP5 era could have caused the increase in the ECS in this model. Apart from increases in horizontal and vertical resolution, the EC-Earth model also has substantially changed the representation of aerosols, and in particular it has introduced a more sophisticated description of aerosol indirect effects. After testing the model with some of the recent updates switched off, we find that the ECS increase can be attributed to the more advanced treatment of aerosols, with the largest contribution coming from the effect of aerosols on cloud microphysics (cloud lifetime or second indirect effect). The increase in climate sensitivity is unrelated to model tuning as all experiments have been performed with the same tuning parameters and only the representation of the aerosol effects has been changing. These results cannot be generalised to other models as their CMIP5 and CMIP6 versions may differ in other aspects than the aerosol-cloud interaction, but the results highlights the strong sensitivity of ECS to the details of the aerosol forcing.

## 1 Introduction

The equilibrium climate sensitivity (ECS) is the average change in global and annual mean near-surface air temperature (T2m) following an instantaneous doubling of the $CO_2$ concentration compared to preindustrial levels, after the climate has reached a new equilibrium. It is a widely used metric in the climate modeling to illustrate the warming from increased $CO_2$ levels including feedbacks in the climate system. The ECS is also highly relevant for climate policy: Matthews et al. (2009) found that global warming mainly depends on the total cumulative anthropogenic emission of carbon to the atmosphere and that the details of the emission pathways are of secondary importance for the warming. The larger the ECS the smaller the

amount of carbon that still can be emitted in order to limit the warming to a value below a given level, e.g. warming levels suggested by the Paris treaty.

Despite the simple definition of the ECS it is not easy to constrain its value, neither with observations nor with models (Roe and Armour, 2011; Knutti et al. 2017). The majority of CMIP5 models have an ECS in the range between 2.1 and 4.7 K (IPCC 2013). With the first results from CMIP6 models becoming accessible, it was found that for a number of models the ECS has increased substantially compared to the values that were found for CMIP5 (Zelinka et al 2020) with the predecessors of the very same models (e.g. Mauritsen et al. 2019, Gettelman et al. 2019, Valdoire et al. 2019), which has already led to discussions about possible implications of higher climate sensitivity (Voosen 2019, https://www.carbonbrief.org/guest-post-why-results-from-the-next-generation-of-climate-models-matter). Our EC-Earth model also shows an increased sensitivity: EC-Earth2 (hereafter ECE2) which has been used for CMIP5 had an ECS of 3.3 K that has increased to 4.3 K in the newer model version EC-Earth3 used for CMIP6 (hereafter ECE3). The goal of this study is to identify and quantify the contributions from model updates when going from ECE2 to ECE3. Unfortunately the complex nature of the model development process makes it impossible to turn back the development steps in a systematic and continuous approach. Some of the newly introduced processes or forcing datasets can only be switched on or off in combination with others, for example the more advanced treatment of aerosol indirect effects can only be used in combination with the new aerosol representation in ECE3 that has nothing corresponding in ECE2. Nevertheless, we attempt to analyse the reasons for the ECS increase with systematic model sensitivity experiments to test the contributions from the various steps during the model developments.

The goal of this study is neither to justify the higher ECS of ECE3 nor that of CMIP6 models in general; we only investigate possible reasons for the increase of the ECS in the EC-Earth model family when advancing from the CMIP5 to the CMIP6 version of the model. General constraints on the ECS are outside the range of this study as well as general findings on the ECS for all CMIP6 models that have been addressed elsewhere (Zelinka et al. 2020, Flynn and Mauritsen 2020). Any conclusion presented here is valid only for the EC-Earth3 model, but since many climate models share model components and/or forcings the findings presented here could hint at possible reasons for higher ECS even in other models.

## 2 Method

### 2.1 The EC-Earth model

The CMIP5 version of the EC-Earth model is based on the ECMWF integrated forecasting system (IFS) cy31r1 and the NEMO version 2 ocean model (OPA9 with the LIM2 sea ice model), see Hazeleger et al (2012) for details. All components have been upgraded for the new EC-Earth3 model that is used for CMIP6. A detailed description of ECE3 is in preparation (Döscher et al., 2020). The differences in model components and resolutions of ECE2 and ECE3 are listed in Table 1. In addition to the differences between model versions there are also differences in the forcing datasets when going from CMIP5

to CMIP6, e.g. the greenhouse gases (GHGs, Meinshausen et al. 2020) or aerosol forcing datasets (Stevens et al 2017) but the impact of the changes in the external forcing on the ECS is outside the scope of this study.

The ECE3 model is used to contribute to CMIP6 in several configurations. For the work here we have used the EC-Earth3-Veg configuration which couples the dynamic vegetation model LPJ-Guess (Smith et al. 2014) to the atmosphere and ocean model, yet the performance of EC-Earth3 and EC-Earth3-Veg is very similar.

A noteworthy difference between ECE2 and ECE3 is the way the aerosols are treated. In EC-Earth2, aerosols are prescribed as mass concentration fields following the CMIP5 time series from the Community Atmosphere Model (CAM, Lamarque et al., 2011). The provided aerosol components are mapped onto the aerosol types used in IFS and fed into the short- and longwave radiation scheme (direct effect). The cloud effective radius $r_{eff}$ is computed by distributing the cloud water on a fixed number of droplets following Martin et al. (1994),

$$r_{eff} = \left( \frac{3L}{4\pi k N_d} \right)^{1/3} \tag{1}$$

where $L$ is the liquid water content, $N_d$ the number of droplets and $k$ a proportionality factor derived from observations. Both $N_d$ and $k$ have fixed values over land and over sea, over land $N$=313.2 and $k$=0.688, and over sea $N$=50.6 and $k$=0.775. The number of droplets is independent of the CMIP5 aerosol data. Hence, ECE2 accounts only for the direct radiative effects of the prescribed changes in aerosol concentrations in the forcing dataset, but has no representation of the indirect effects via the aerosol impact on clouds.

ECE3 includes a representation of the direct and indirect aerosol effects. For the direct aerosol effects in the shortwave the model uses the optical properties of the aerosol plumes provided by the MACv2-SP simple plume model (Stevens et al. 2017) in combination with monthly climatologies of the optical properties of the pre-industrial background aerosol concentration that have been obtained from an off-line simulation using the atmospheric composition model TM5 (Van Noije et al. 2014; Bergman et al. 2020) forced with pre-industrial emissions for CMIP6 (Hoesly et al., 2018; Van Marle et al., 2017). The aerosol effects in the longwave are calculated based on the background aerosol mass concentrations obtained from the pre-industrial TM5 simulation. For the indirect aerosol effect the number of activated aerosols is computed following the work of Abdul-Razzak and Ghan (2000). The Abdul-Razzak and Ghan scheme parameterises the number of activated aerosols of multiple externally mixed lognormal aerosol modes, each composed of a uniform internal mixture of soluble and insoluble material. The Köhler theory is used to relate the aerosol size distribution and composition to the number activated as a function of maximum supersaturation. The supersaturation balance is used to determine the maximum supersaturation, accounting for particle growth both before and after the particles are activated.

Indirect aerosol effects are accounted for by making the effective radius and autoconversion efficiency depend on the concentration of cloud droplets (CDNC). The effective radius is still computed as in (1) yet with the constant droplet concentration $N_d$ replaced by the dynamic CDNC. The autoconversion efficiency $a$ is a linear function of cloud water above a given threshold following Sundqvist (1978):

$$a = c_0 \left( 1 - e^{-\left(\frac{L}{L_{crit}}\right)^2} \right) \qquad (2)$$

where $c_0^{-1}$ represents a characteristic time scale for the conversion of cloud liquid droplets, $L$ is the liquid water content and $L_{crit}$ the typical cloud water content at which the generation of precipitation begins to be efficient. To account for the variation in the cloud droplet number through aerosol activation, the autoconversion efficiency is scaled with CDNC following Rotstayn and Penner (2001):

$$a' = a \left( \frac{N_0}{N} \right)^{1/3}$$

with $N_0$=125 cm$^{-1}$ a typical and $N$=CDNC the actual cloud condensation nuclei concentration.

The aerosol number and mass concentration fields that serve as input to the activation scheme are climatologies from the off-line pre-industrial run with TM5. The changes in aerosol concentrations since the pre-industrial era in transient simulations for CMIP6 are accounted for by multiplying the resulting cloud droplet number concentration by the multiplication factor provided by MACv2-SP. Note however that in the piControl and abrupt-4xCO2 experiments for this study the pre-industrial aerosol concentrations are used without any multiplication factor, following the experimental protocol for these CMIP6 experiments.

## 2.2 Experiment design

ECS is assessed by comparing the response of the net top-of-the-atmosphere (TOA) radiative flux ($Q_{net}$) and T2m from the abrupt-4xCO2 experiment (hereafter denoted as 4xCO2) against the steady climate of the pre-industrial control experiment (piControl) with its baseline $CO_2$ concentration. Each model modification therefore requires two long model simulations, one with baseline and one with quadrupled $CO_2$ concentration. In a first step we compare ECS between the CMIP5 and CMIP6 version of the EC-Earth model. We then analyse changes in the global means and in the regional distribution of clouds and their impact on the cloud forcing (CF, the difference between all-sky and clear-sky net radiation) to investigate the difference in climate sensitivity between ECE2 and ECE3.

To better understand the role of the various improvements during the development process of ECE3 we roll back some of the changes and measure the impact on CF and ECS in a series of sensitivity experiments. Apart from the changes in model resolution, the most relevant updates of the model are those related to the revised treatment of the aerosols and their interaction with clouds. The question is if and possibly how much any of these changes have contributed to the increase in ECS that we find when comparing the CMIP5 and the CMIP6 version of the EC-Earth model.

The CMIP6 protocol requires the 4xCO2 experiments to be 150 years long, but in order to save computational resources we test if sensitivity experiments of only 75 years length could give an acceptable estimate for the ECS. In another attempt to save computational resources we investigate if the ECS depends on the model resolution. The horizontal and vertical resolution of the atmosphere model in EC-Earth3 is reduced to the resolution that was used for CMIP5. A reduction of the

simulation length and the lower resolution allows us to perform more experiments with the available computational resources but of course we first need to establish that these modifications have only a small impact on the ECS of the model.

## 2.3 Assessing the equilibrium climate sensitivity

ECS is defined as the increase in the global mean T2m between a steady-state climate with pre-industrial levels of CO2
concentrations and the steady-state climate with doubled $CO_2$ concentrations, with all other forcings (GHGs, aerosols, land-use etc.) remaining at pre-industrial conditions. Despite this simple and straightforward definition of the ECS the practical task to assess the ECS of a model is a real challenge because it would require the model to run with increased $CO_2$ concentration until it reaches equilibrium. However, the brute force approach to run the model until equilibrium is not very practical as it would take thousands of years of model integration to bring the deep ocean into equilibrium and to find the steady-state equilibrium temperature (e.g. Stouffer 2004, Paynter et al 2018, Rugenstein et al. 2020). For this reason modellers often apply the method proposed by Gregory et al. (2004) that also has been used to estimate ECS in CMIP5 (IPCC 2013, Andrews et al. 2012) and CMIP6 models (e.g. Andrews et al. 2019, Voldoire et al. 2019, Zelinka et al. 2020). Here we apply the Gregory method to the 4xCO2 experiments for CMIP6 which are only 150 years long (Eyring et al. 2016). When doing a simulation with increased $CO_2$ concentrations the global mean $Q_{net}$ and T2m asymptotically approach the equilibrium state, and by extrapolating a linear fit of the data points to the $Q_{net}=0$ level one can obtain an estimate of the equilibrium temperature that would be reached when the model reaches its new equilibrium that is characterized by a zero TOA energy balance. Apart from ECS the Gregory method allows one to estimate two other important model: the intercept of the linear fit the ordinate indicates the effective radiative forcing (ERF) for a quadrupling of the $CO_2$ concentration, while the slope of the linear regression is known as radiative feedback parameter ($\lambda$) that expresses the strength of the feedback. Since models may present a not perfectly closed energy balance, resulting in a non-zero equilibrium TOA net flux, the preindustrial equilibrium values are typically removed from the 4xCO2 values before proceeding with the fit to determine ECS.

ECS by definition is the temperature change that results from a doubling of the $CO_2$ concentrations. However, the DECK (Diagnostic, Evaluation and Characterization of Klima) experiments for CMIP6 comprise the abrupt4xCO2 experiment with instantaneously quadrupled $CO_2$ (Eyring et al., 2016). Following common practice (e.g. Andrews et al 2012, IPCC 2013, Knutti et al. 2017, Zelinka et al. 2020) we divide the estimate for the equilibrium temperature and effective radiative forcing in the 4xCO2 experiment by 2 to obtain estimates for ECS and ERF for a doubling of the CO2 concentration.

### 2.3.1 Correction for model drift

With a steady state control climate in the piControl experiment it is straightforward to evaluate the TOA radiation imbalance and temperature response at the surface in a sensitivity experiment with changed forcing relative to the control climate. The control climate and response to changed forcing are evaluated in corresponding time periods in the control and sensitivity experiment, respectively. However, when testing the sensitivity of the ECS to recent model changes we switch on/off some

model features, which may result in an ill-tuned model and introduce a drift. In principle one would have to first make a new spin-up run with the modified model before starting new piControl and 4xCO2 experiments, yet limited computational resources don't allow us to make several long spin-up runs with slightly modified model configurations. To overcome this difficulty we assume that the model modifications lead only to a small drift in the pre-industrial control climate that we can correct for. After making the experiment with pre-industrial forcing with each model modification we first make linear fits of the $Q_{net}$ and T2m time series and then use these regressions to correct the time series of the corresponding 4xCO2 experiment (Fig. 1), following common practice (e.g. Andrews et al., 2015). We applied a similar correction also to the unperturbed control experiment. Since the largest shock caused by a model modification occurs right at the start of the simulation and may give rise to a non-linear response we exclude the first 5 annual means when computing the linear fit for the model drift. For the same reason we also exclude the first 5 years of the net radiation and temperature time series when computing the linear regression for estimating the ECS. We have tested the impact on the ECS when excluding a few years from the data, and find that the result doesn't change any longer if 4 or more years are excluded.  We have verified that the resulting ECS estimates are very close to the values obtained with more advanced linear regression methods that are more robust against outliers (e.g. Theil-Sen regression), confirming that the strongest deviations from the linear relation are indeed observed during the first few years.

## 3 Results

### 3.1 Climate sensitivity in ECE2 and ECE3

Table 2 presents the ECS, net feedback and ERF for the CMIP5 and CMIP6 version of the EC-Earth model. ECS increases from 3.34 K to 4.31 K. Zelinka et al (2020) conclude that the higher ECS of many CMIP6 models is due to a combination of higher ERF and weaker net feedback compared to the model versions that have been used for CMIP5. However, the EC-Earth model is slightly different compared to other models because its ERF doesn't change much between the CMIP5 and CMIP6 version and therefore the different ECS in ECE2 and ECE3 is mainly caused by a different net feedback parameter. The small change of ERF from ECE2 to ECE3 can explain the comparably weak increase of ECS in the EC-Earth model, other models show considerably larger increase between their CMIP5 and CMIP6 versions yet Zelinka et al. (2020) conclude that the differences are not significant.

To analyse the causes for the change in the net feedback parameter look at the response in the 4xCO2 experiments in ECE2 and ECE3. Fig. 2 shows the change in clouds at the end (years 131 to 150) of the in the 4xCO2 experiment relative to the piControl experiment. We find a different behaviour in the versions of the model: ECE2 shows a weaker response in the cloud cover than ECE3, in particular over North Hemisphere Atlantic and Pacific Ocean, while ECE2 shows a stronger response in cloud liquid water path (LWP) in the extratropics. The response of the vertically integrated cloud ice has a similar pattern in ECE2 and ECE3 but is somewhat stronger in ECE3 (not shown). These differences in the response of the cloud fraction and LWP due to a quadrupling of CO2 have also an impact on the cloud forcing (Fig. 3). In ECE2 the

response in the CF is weak except at high latitudes which results from the melting of sea-ice in a warmer climate. In contrast ECE3 shows a more pronounced response in the cloud forcing. In the tropics CF becomes more negative and over the Northern Hemisphere Atlantic and Pacific Ocean the response is positive leading to a less negative cloud forcing.

    These changes in the response of clouds and subsequently cloud forcing can explain the change in the climate sensitivity of the EC-Earth model when going from the CMIP5 version to the CMIP6 version. The question is then what modifications of 190     the cloud parameterisation during the development of ECE3 play an important role for the changes in the response to an increased $CO_2$ forcing, and what impact these model updates have on the ECS. To study the effects from different model development steps, we roll back the developments that are related to the aerosol and cloud interaction, and then repeat the piControl and abrupt4xCO2 experiments in a series of sensitivity studies.

### 3.2 Reducing the length of the simulation

Reducing the length of the piControl and 4xCO2 simulations would make the sensitivity experiments computationally cheaper, but it could only be done if the impact on the ECS is small. In order to test this, we compute the ECS from our DECK experiments (EC-Earth Consortium 2019) by taking 150 and of 75 years of the annual mean timeseries, respectively. In both cases the model configuration is EC-Earth3-Veg with the full T255L91-ORCA1L75 resolution used for CMIP6. The ECS is found to be not significantly different irrespective of including 150 or 75 years in the linear regression (Table 2). We 200     therefore conclude that we can safely reduce the length of the sensitivity experiments with minimal impact on the ECS.

### 3.3 Reducing the model resolution

    In another attempt to reduce the computational costs of the sensitivity simulations we test if the horizontal and vertical resolution of ECE3 could be reduced to that of ECE2 (Table 3). In these tests we have changed the resolution (only in the atmosphere) first only in the horizontal and then in both horizontal and vertical. The ECS changes slightly from 4.3 K to 4.2 205     or stays close to 4.3 K when only the horizontal or both the horizontal and vertical resolution has been changed.  These changes in ECS are small compared to the difference in ECS between ECE2 and ECE3. An important result of these tests is that the change in the resolution of the EC-Earth model from CMIP5 to CMIP6 is not responsible for the change in the climate sensitivity and the reasons have to be sought elsewhere, Because the changes in the resolution don't have only a marginal impact on the ECS the sensitivity experiments with modified aerosol-cloud interaction are made with the low 210     resolution configuration of EC-Earth3-Veg. The resulting ECS will not be fully accurate for the full-resolution CMIP6 model; nevertheless the estimates obtained with low resolution will allow us to make a qualitative assessment of the impact of the newly implemented aerosol scheme.

### 3.4 Sensitivity to the description of aerosols and their impacts on the cloud forcing

    The results from a series of sensitivity experiment with the aerosol scheme in ECE3 are shown in Table 4. When reverting 215     the newly implemented simple plume representation of MACv2-SP in combination with a pre-industrial background

climatology back to the scheme with prescribed aerosol concentrations used for CMIP5, we find that the ECS drops to 3.25 K which is close to the value that was found for the CMIP5 version of EC-Earth. Changing the source of the aerosol forcing from the CMIP5 data set to the new representation of aerosol optical properties in CMIP6 but without aerosol indirect effects - effective radius and autoconversion are parameterised as in the CMIP5 version of the model and do not depend on the

220 number of activated aerosol particles calculated from the pre-industrial climatology of aerosol concentrations - the ECS increases slightly to 3.54 K. The change is small and not significant with all the simplifications of the experimental design in mind. When the coupling between the explicit aerosol activation is switched on and impacts the effective radius (1st indirect effect) the ECS increases further to 3.81 K, and if in addition the activated aerosol particles are also allowed to impact cloud microphysics the ECS becomes 4.28 K. This last value is similar to the ECS from the CMIP6 experiments (4.31 K) with EC-

225 Earth3-Veg performed at higher atmospheric resolution (T255L91).

This series of sensitivity experiments suggests that the increase of the ECS from CMIP5 to CMIP6 is mainly caused by the change in the representation of aerosol and their impacts on clouds and radiation. The implementation of MACv2-SP as it is suggested for CMIP6 models without explicit aerosol scheme has fundamentally changed the way how aerosols are prescribed in the model, yet this change has little effect on the ECS as long as cloud droplet effective radius and

230 autoconversion are independent of the aerosol concentration. The ECS increases when the more advanced treatment of the first and second indirect effect is introduced, with the largest contribution coming from the latter. This finding is further supported by the change in the net CF in these sensitivity experiments. The largest change in the CF is found when the 2nd indirect aerosol effect is activated. In that case the cooling by clouds becomes less strong (CF increases) which reduces the feedback from the warming induced by the quadrupled $CO_2$ concentrations resulting in a higher ECS.

Kiehl (2007) has shown a correlation between stronger aerosol forcing and higher climate sensitivity in climate models. By introducing a more advanced treatment of aerosols in the EC-Earth3 model and subsequent tuning to match a realistic preindustrial equilibrium and present-day climate in the model we may have altered the model's sensitivity. However, we have shown here that the ECS can also change without changing the model tuning. It is possible to get back the climate sensitivity of ECE2 with the new ECE3 model when reverting the changes in the representation of the aerosol cloud

interaction without any retuning of ECE3.

## 4 Conclusions

The ECS of the EC-Earth model has increased from 3.3 K in CMIP5 to 4.3 K in CMIP6. In this work we show that this increase can be explained by the revised description of aerosol processes when going from ECE2 to ECE3, in particular the

245 implementation of the first and second indirect aerosol effect. In fact, cloud feedbacks have been identified among the most important sources of uncertainty for ECS for the past generation of climate models (Andrews et al. 2012). Interestingly the

analysis by Chylek et al. (2016) suggested that only CMIP5 models including indirect aerosol effects present a correlation between radiative forcing and equilibrium climate sensitivity similar to that discussed in Kiehl (2007).

Of course, the question has to be asked how good is the representation in EC-Earth3 of specific processes such as the activation of aerosols, how realistic are the parameterisations of effective radius and autoconversion efficiency as a function of the activated cloud droplets, and how will all these changes affect the ECS of the model. The coming CMIP6 experiments in AerChemMIP will help us to better understand how well the EC-Earth3 model represents such aerosol-cloud interactions. All results from this study are valid for the EC-Earth3 model only. Many of the other climate models already had indirect aerosol effects in their CMIP5 version and therefore they cannot easily explain an increase of the ECS with the introduction of a more sophisticated aerosol scheme. However, many models have updated their aerosol representation since CMIP5 and some have implemented the new MACv2-SP scheme. It is possible – but impossible to prove here – that the changes in the aerosol treatment could make a substantial contribution to the increase in ECS that many modelling groups have found.

The development of the next generation of the EC-Earth model has already begun. One of the lessons learned from the large increase in ECS when going from the CMIP5 to the CMIP6 version of the model is that we will have to carefully monitor the climate sensitivity (and other key metrics) not only at the end but also during the entire development process as it was done for example for the CESM2 model (Gettelman et al. 2019). Maintaining a well-tuned model version and at the same time having a continuous picture of the ECS evolution and of the main feedback parameters over time will support us in the critical evaluation of any new model developments and suggest a critical re-tuning the model whenever important changes in climate sensitivity are found. .

**Code and data availability**

The EC-Earth model is restricted to institutes that have signed a memorandum of understanding or letter of intent with the EC-Earth consortium and a software license agreement with ECMWF. Confidential access to the code can be granted for editor and reviewers, please use the contact form at http://www.ec-earth.org/about/contact. The data from the piControl and abrupt4xCO2 for CMIP5 are available from https://doi.org/10.5281/zenodo.3459914 while the CMIP6 data can be downloaded from any ESGF dataportal (cf. reference EC-Earth Consortium 2019). The results of the sensitivity experiments with EC-Earth3-Veg used in this study are available from https://doi.org/10.5281/zenodo.3454079.

**Author contribution**

All co-authors are part of the EC-Earth consortium that develops the EC-Earth model. The experiments with EC-Earth3 were done by K. Wyser while the experiments with EC-Earth2 were done by S. Yang. All co-authors have participated in the analysis of the results. K. Wyser prepared the manuscript with contributions from all co-authors.

## Acknowledgement

This work was supported by the European Union's Horizon 2020 research and innovation programme under grant agreement No 641816 (CRESCENDO). The EC-Earth simulations were performed on resources provided by the Swedish National Infrastructure for Computing (SNIC) at PDC and NSC.

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

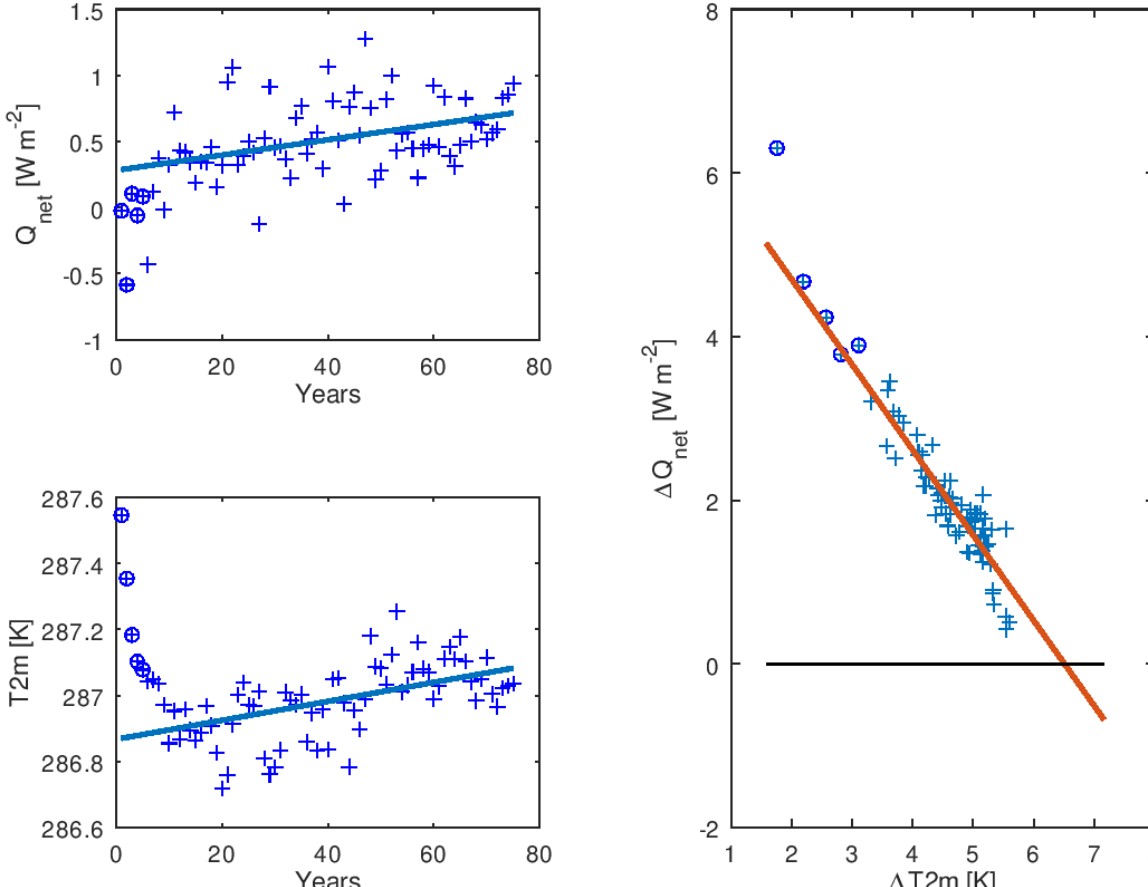

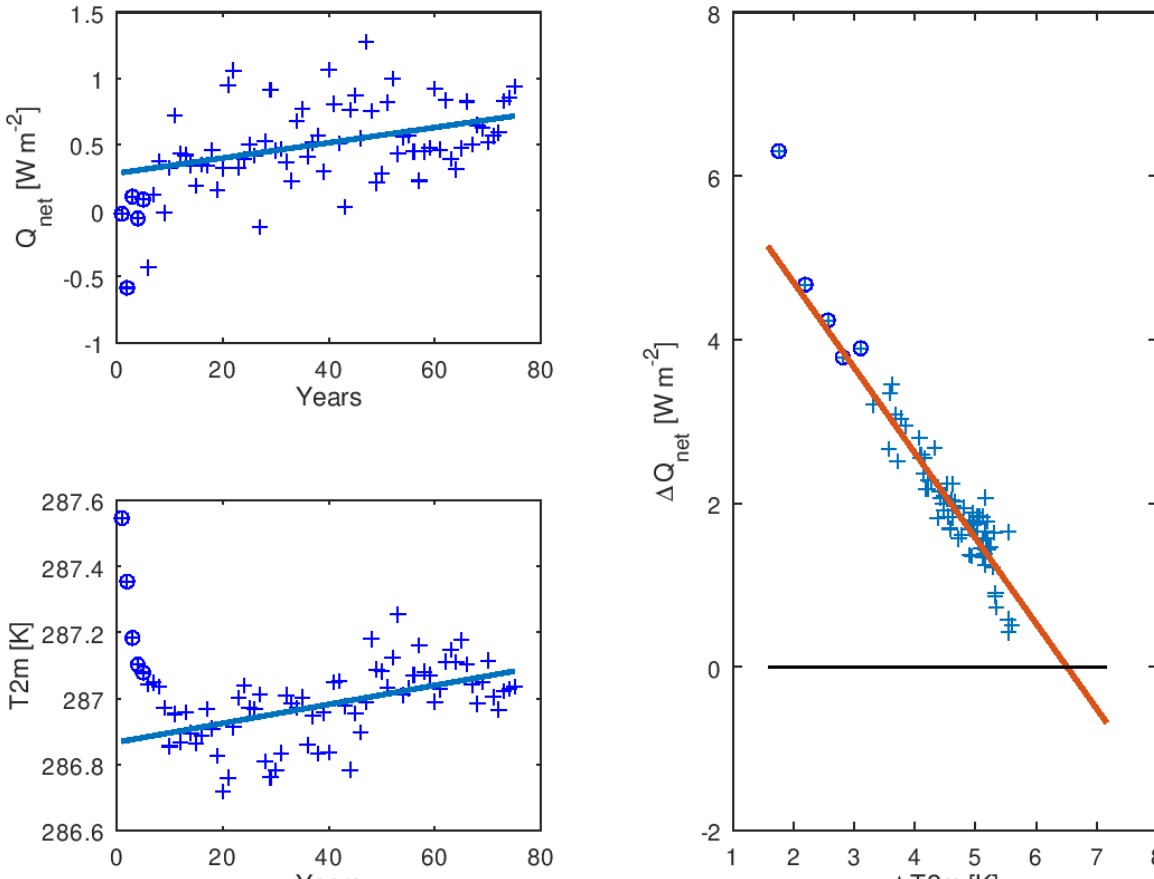

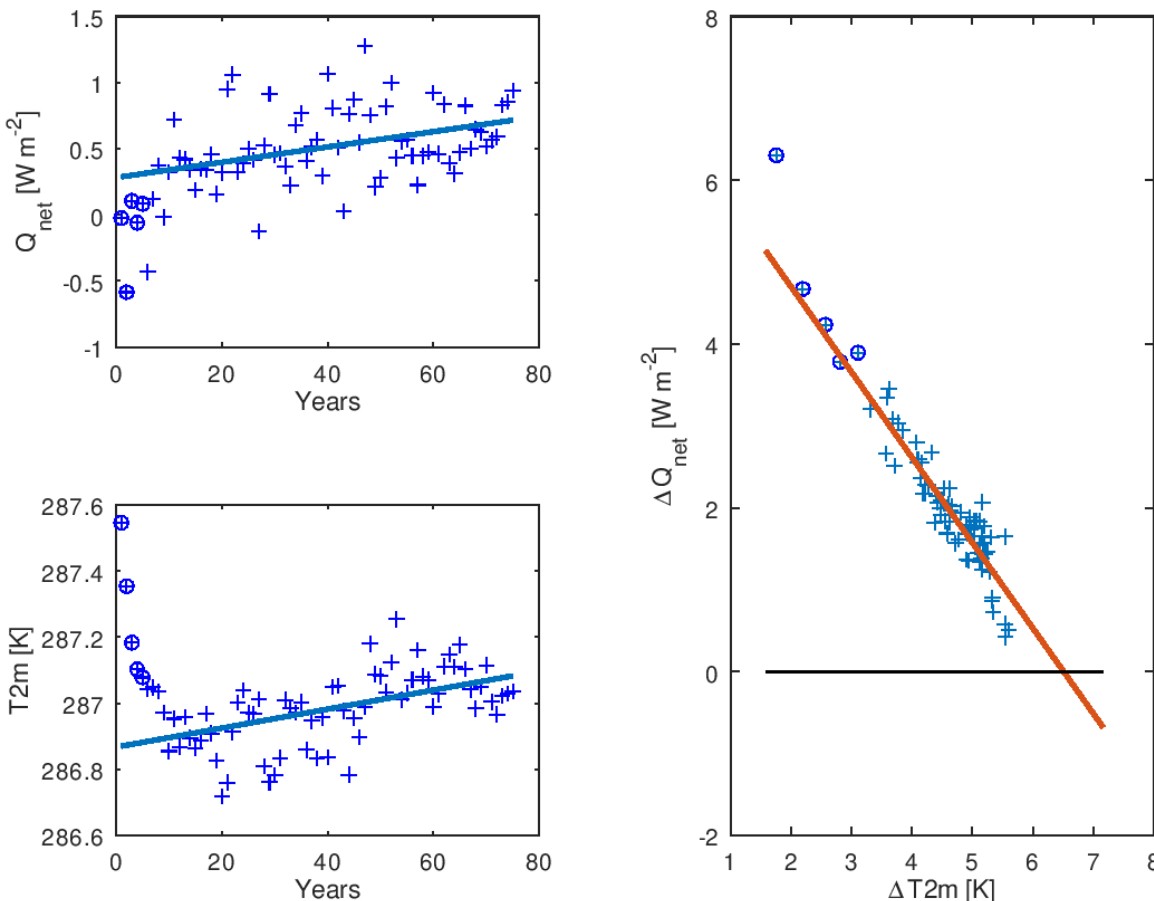

**Figure 1, left: Timeseries of $Q_{net}$ and T2m in a pre-industrial simulation with CMIP5 aerosols and without explicit cloud droplet activation. The model isn't tuned for this configuration and therefore experiences a drift over time. The linear regression (solid) in the $Q_{net}$ and T2m plot provides the offset and drift correction- that are later subtracted from the 4xCO2 experiment with the same model configuration. The first 5 years (marked with "o" in the plot) are excluded when computing the linear fit. Right: Gregory plot from the 4xCO2 experiment for the same model configuration after correcting for offset and drift in the corresponding experiment with pre-industrial forcing. A regression line is fitted to the data points (red) and extrapolated, again excluding the first 5 years marked "o"). The intersection of this line with the $\Delta Q_{net}$=0 line is an estimate for the equilibrium temperature response in the 4xCO2 experiment. This value has to be divided by 2 to yield an estimate for the ECS.**

**EC-EARTH v2.3**

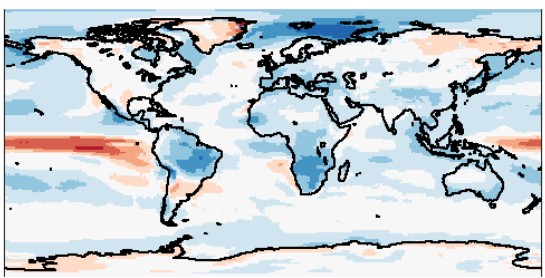

**EC-EARTH v2.3**

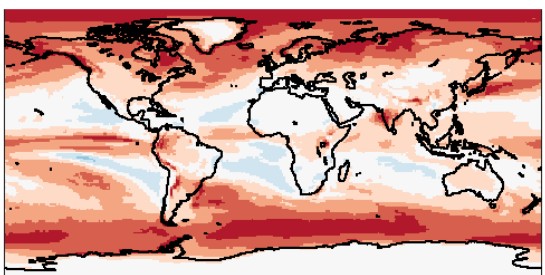

**EC-Earth3-Veg**

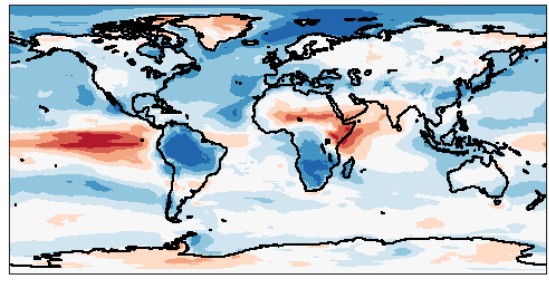

**EC-Earth3-Veg**

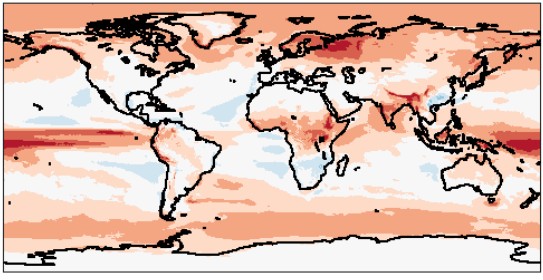

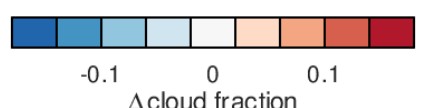

$\Delta$ cloud fraction

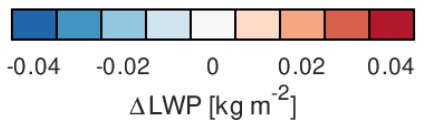

$\Delta$ LWP [kg m$^{-2}$]

**EC-EARTH v2.3**

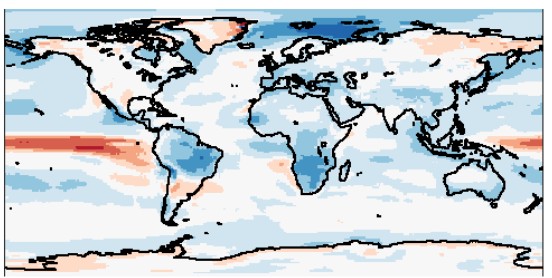

**EC-EARTH v2.3**

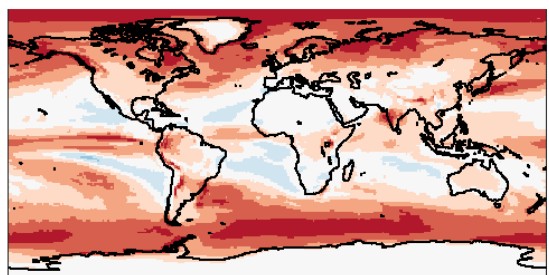

**EC-Earth3-Veg**

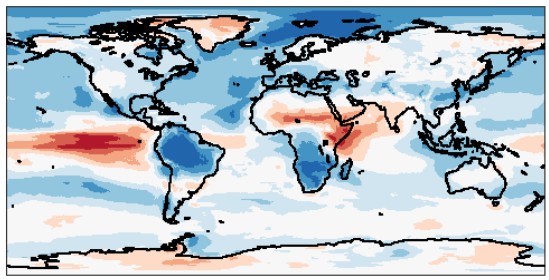

**EC-Earth3-Veg**

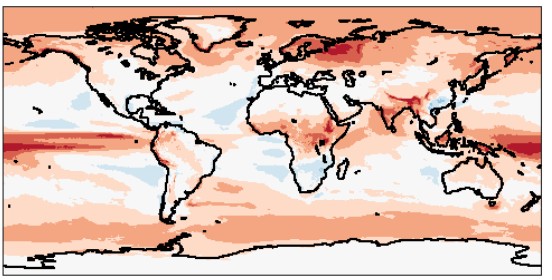

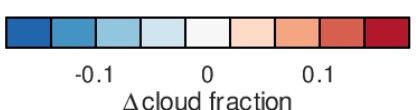

$\Delta$cloud fraction

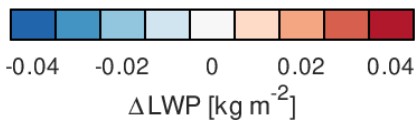

$\Delta$LWP [kg m$^{-2}$]

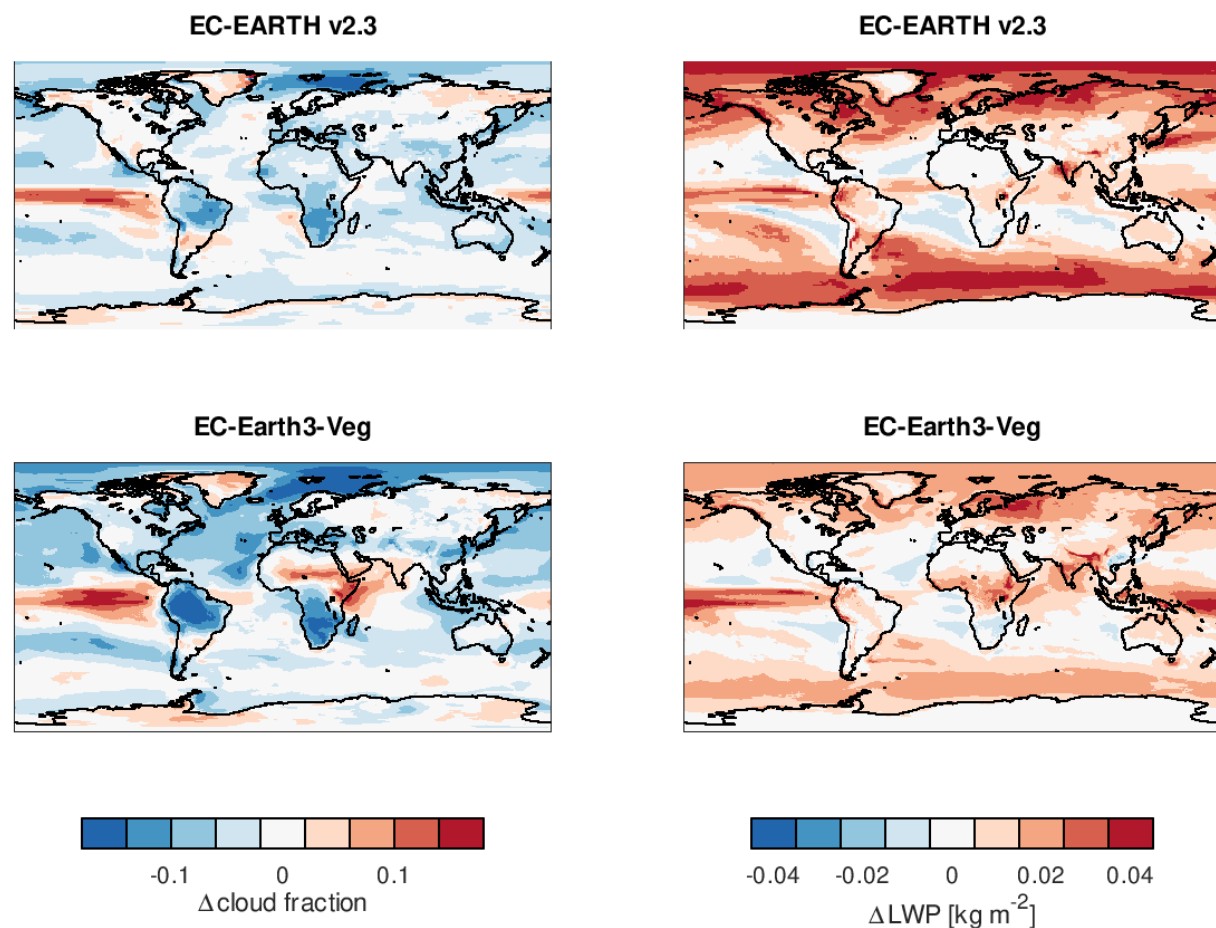

**Figure 2. Response of cloud fraction (left) and LWP (right) to a quadrupling of CO2 in ECE2 (top) and ECE3 (bottom).**

**EC-EARTH v2.3**

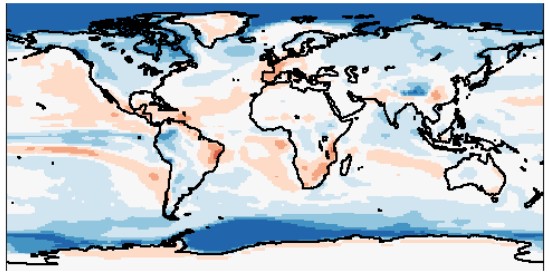

**EC-Earth3-Veg**

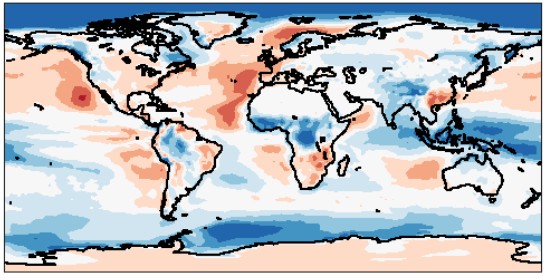

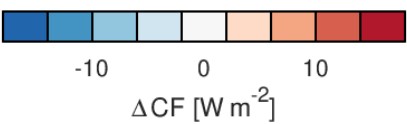

$\triangle$CF [W m$^{-2}$]

**EC-EARTH v2.3**

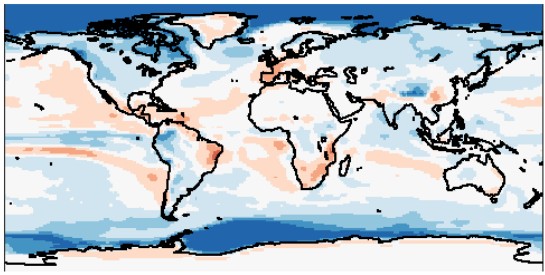

**EC-Earth3-Veg**

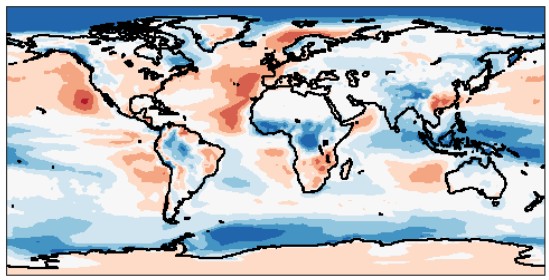

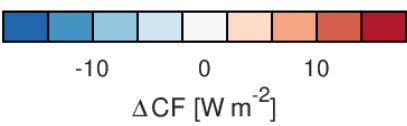

$\triangle CF \ [W \ m^{-2}]$

**EC-EARTH v2.3**

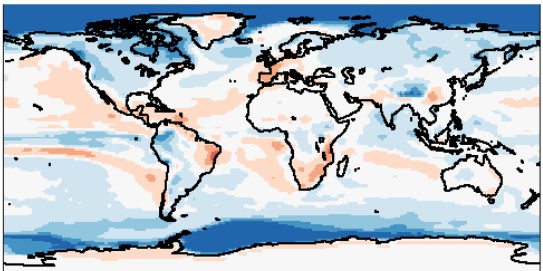

**EC-Earth3-Veg**

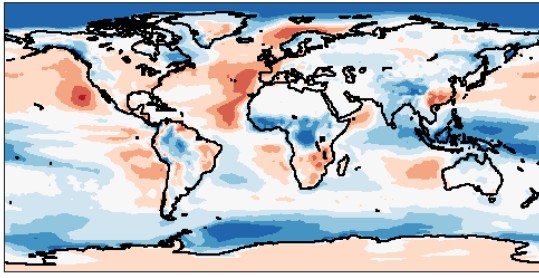

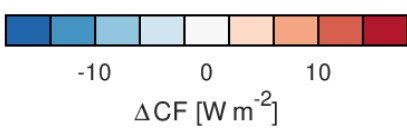

**Figure 3. As Fig. 2 but for net cloud forcing.**

| | EC-Earth2 (ECE2) | EC-Earth3 (ECE3) |
|---|---|---|
| Atmosphere model | IFS cy31r1 | IFS cy36r4 |
| Ocean model | NEMO2 (OPA9) | NEMO 3.6 |
| Sea-ice model | LIM2 with 1 sea ice category | LIM3 with 5 sea ice categories |
| Resolution — Atmosphere | T159L62 (125 km) | T255L91 (80 km) |
| Resolution — Ocean | ORCA1L42 (1 deg) | ORCA1L75 (1 deg) |

**Table 1: CMIP5 and CMIP6 versions of the EC-Earth model family**

| MIP | Model | ECS | λ | ERF |
|---|---|---|---|---|
| CMIP5 | EC-Earth2 | 3.34 ± 0.05 | -1.01 ± 0.03 | 3.37 ± 0.13 |
| CMIP6 | EC-Earth3-Veg | 4.31 ± 0.08 | -0.79 ± 0.03 | 3.41 ± 0.17 |

**Table 2: Equilibrium climate sensitivity (ECS in K), net feedback parameter (λ in W m$^{-2}$ K$^{-1}$), and effective radiative forcing (ERF in W m$^{-2}$) in the CMIP5 and CMIP6 version of the EC-Earth model**

| Model | Length (years) | Resolution | ECS | Remarks |
|---|---|---|---|---|
| EC-Earth2 | 150 | T159L62-ORCA1L42 | 3.34 ± 0.05 | Used in CMIP5 |
| EC-Earth3-Veg | 150 | T255L91-ORCA1L75 | 4.31 ± 0.08 | Used in CMIP6 |
| | 75 | T255L91-ORCA1L75 | 4.27 ± 0.15 | Reduced length |
| | | T159L91-ORCA1L75 | 4.03 ± 0.12 | Reduced length + reduced horizontal resolution |
| | | T159L62-ORCA1L75 | 4.11 ± 0.12 | Reduced length + reduced horizontal and reduced vertical resolution |

**Table 3: Impact of a reduced simulation length and reduced model resolution on the ECS. The ECS value for EC-Earth2 is shown for comparison.**

| Experiment | Aerosol direct radiative effect | First indirect effect | Second indirect effect | ECS (K) | Net CF (W m$^{-2}$) |
|---|---|---|---|---|---|

| Prescribed aerosol concentrations from CMIP5 | As for CMIP5 | As for CMIP5 | As for CMIP5 | 3.25 ± 0.07 | -21.54 ± 0.32 |
|---|---|---|---|---|---|
| Aerosols as in CMIP6 | As for CMIP6 | As for CMIP5 | As for CMIP5 | 3.54 ± 0.12 | -21.31 ± 0.34 |
| | As for CMIP6 | As for CMIP6 | As for CMIP5 | 3.81 ± 0.12 | -21.69 ± 0.26 |
| | As for CMIP6 | As for CMIP6 | As for CMIP6 | 4.28 ± 0.12 | -18.07 ± 0.28 |

**Table 4: Sensitivity of ECS and net CF to different realisations of the aerosol-cloud interaction processes. All experiments were done with the low resolution (T159L62) configuration of EC-Earth3-Veg and stretch over only 75 years.**