# Peer review of "On the increased climate sensitivity in the EC-Earth model from CMIP5 to CMIP6"

_Geoscientific Model Development, 2019_

## Referee Comment (RC1) · Anonymous Referee #1 · 29 Jan 2020

In the paper the authors discuss the estimated Equilibrium Climate Sensitivity (ECS) from 11 or so simulations with different set ups of the EC-Earth model. They conclude that the 1K increase of ECS from CMIP5 to CMIP6 stems from the implementation of the first and second indirect aerosol effect. Increasing the vertical and horizontal resolution, including interactive vegetation, and new model tuning play a minor or no role.

The content of the paper is very relevant in the current discussion on reasons of higher ECS in many CMIP6 models and the simulations are well justified and documented in the tables. However, the paper seems pretty hastily written, references to the literature are missing, and the analysis is very superficial (global mean only with no discussion on actual reasons and local expression of differences), compared to any scientific paper

and also to papers documenting high sensitivity of other models (e.g. Gettleman et al. 2020 and Zelinka et al. 2020)

Major comments

1) The paper could be easily improved by describing the changes to the model code in terms which can be understood by non-aerosol specialists, e.g. somebody who might be interested model weighting and query your paper for understanding why EC-Earth model version X has a higher sensitivity that EC-Earth model version Y; which of the two to include in a certain assessment and how much weight to give each model version. Such a person should be able to take more away from your paper than "… [the change] is mainly caused by the change in the representation of aerosols and their impact on clouds and radiation" (line 179), isn't it? It would help to flesh out line 65-88 (not necessarily in length but in readability for non-experts).

2) I my eyes, a main conclusion of the paper is "This series of sensitivity experiments suggests that the increase of the ECS from CMIP5 to CMIP6 is mainly caused by the change in the representation of aerosol and their impacts on clouds and radiation. The implementation of MACv2-SP as it is suggested for CMIP6 models without explicit aerosol scheme has fundamentally changed the way how aerosols are prescribed in the model, yet this change has little effect on the ECS as long as cloud effective radius and autoconversion are independent of the aerosol concentration. The ECS increases when the more advanced treatment of the first and second indirect effect is introduced, with the largest contribution coming from the latter" (line 177-184 or so)

The paper would profit from putting actual numbers behind these claims, which are now only expressed in Table 1 and 2 (the only actual results of the paper). Simply showing the relative contributions locally, some spatial fields of how the second indirect effect is expressed, and the different contributions (short and long wave, clear and full sky) to Delta Q net (as in Fig. 1), which is standard in climate sensitivity papers, e.g. Andrews et al. 2012 or 2015, or Zelinka et al. 2020 "Causes of higher climate

sensitivity in CMIP6 models" (which goes much deeper in the discussion but does not replace papers like this one). However, there should be more depth than "it's likely the second indirect effect".

Why? Where? How trustworthy is it (which model versions should I use for which purpose)? Why did you include it into the model? What do we learn from it about climate sensitivity and projections of warming (and precipitation) in the next century? With the men power of six co-authors you should be able to go into a little bit more depth (e.g. compare Gettleman 2019 "High Climate Sensitivity in the Community Earth System Model Version 2 (CESM2)".

3) The paper discusses changes in ECS of 1K and smaller changes between the model versions. There should be uncertainty estimates for all ECS estimate and a discussion in how far differences are even statistically detectable (I'm guessing the difference in line 155 between 4.3 and 4.2K is not even significant (as you also argue, but you should show it)).

Minor comments

line 21: delete "easily" (of course this can't be generalized to any other model (?)) line 27: what's "the climate change context" (?) line 30: ... for the warming. I guess in the next century? Are there more recent papers on this discussion? line 34: "was found" and "has been found" sounds as if these models fell from the sky onto your desk. The models were made more sensitivity (by people like you, changing the model), this is active human action and doesn't passively happen. line 40: "An important question is if we understand the reason for this increase" This statement is very sadly showing the level of climate science these days. I hope you are able to understand the reason for this increase and both the user and the wider public want to know it. If you don't understand the reason, why should I trust you, your model, the IPCC assessment using the model, ... ? line 43: "Unfortunately the complex nature of the model development process makes it impossible to turn back the development in a systematic and continuous approach." What about starting systematically in the first place (!?) *You* developed the model and made these decisions only a few years or even months ago. I know this is a hard process, but the necessity for careful documentation has been openly discussed now for several years (e.g. Hourdin et al., 2016, "The art and science of climate model tuning.", but also already Mauritsen et al. 2012 "Tuning the climate of a global model"). You at least have to blame yourself (or colleagues) for following this approach and not take it as passively given. line 56: This is the last time I'm making that comment: "The EC-Earth global climate model has evolved from ..." You did evolve the model (actively) and should be able to confidently explain me why things change, isn't it? line 71: Explain how (what is in Martin 1994). line 71-73: More explanation and depth is necessary for a reader not familiar with the model or aerosols in general. line 75: Again more explanation necessary. How are these plumes pre-scribed? Do they change in time in the historical but not in the step forcing (the same way as in the control simulation)? A non-aerosol expert would profit here from some plots visualizing the changes in the model. line 79: the *direct* aerosol effect? What's the background aerosol mass concentration? line 81: shortly explain effective radius, auto conversion-efficiency, activation scheme in laymen's terms line 85: ... are ac-counted for by multiplying the resulting cloud droplet number ... I couldn't figure out to what "resulting" refers to. Resulting from what? line 91: What's the baseline $CO_2$ concentration? Is it the same in all models in CMIP6? I think that wasn't the case for CMIP5? How did you deal with this in your model development? line 126: No that's not a basic assumption of the Gregory method. It is very common to detrend models, e.g. Proistosescu and Huybers 2017 or Andrews et al. 2015. (rest of the section is fine though) line 139: In how far does discarding the first five years change the results? See comments on uncertainty above. line 169: Does the lower value refer to the fixed vegetation version? line 181: as long as cloud *droplet(?)* effective radius ... line 184: What's the reason for the correlation in the Kiel 2007 paper? Are there any updates of the discussion in the literature? line 183: This is great that the tuning is discussed! line 189: add maps of these quantities or short wave cloud radiative effects or other feedbacks here (?) line 205: delete "strictly speaking" (or explain how "loosely speaking" your results are valid for any other model (of course they are not (?))) line 221: "All co-authors have participated in the analysis of the results" I see that the model development and running is hard work but I don't see how the analysis takes six people as only single ECS numbers are produced which is a straightforward and standard task. There are no actual analyses done in the paper. line 298: give panels names a, b, c line 310: Express resolution also in approximate km units

References to statements missing throughout the text (e.g. line 62, whole section 2.3, especially line 110, line 113, line 124, and other places)

---

## Referee Comment (RC2) · Anonymous Referee #2 · 5 Feb 2020

The authors present climate sensitivity estimates for different versions of the EC-Earth climate model and attribute the in comparison to the CMIP5 version higher climate sensitivity in the CMIP6 version to the use of a new aerosol climatology and the introduction of aerosol indirect effects. Climate sensitivity is a key metric to describe the global scale response of climate models to CO2 increase. Therefore, understanding reasons for changes is very important and an absolutely appropriate topic for publication in GMD. This is the more the case as several CMIP6 models show higher climate sensitivities than their predecessor and it is important to understand if that is for similar or very diverse reasons and what can be learned from this. Unfortunately, the analysis presented in the paper seems very superficial to me. I think major modifications will be needed before the paper could be acceptable for publication in GMD. In the following I

will list three major concerns followed by a list of further minor comments.

The authors present climate sensitivity estimates from a series of simulations including all or subsets of the aerosol related changes between the CMIP5 and CMIP6 versions. Simulation results indicate that as well direct aerosol effects from the new climatology as the inclusion of different indirect aerosol effects contribute to the higher climate sensitivity. However, no attempt is made to understand why and how these effects come about, which I think would be crucial to make this paper useful for readers beyond the notion that differences in climate sensitivities simulated by different EC-Earth version may be related to the treatment of aerosols. There should be an attempt to attribute the changes to different types of climate feedbacks, as well as to identify possible regional patterns and mechanisms that affect the feedbacks. One could e.g. imagine that the modifications affect ECS rather indirectly, e.g. by modifying the climatological cloud distribution which then reacts differently to an increased GHG concentration. It would be important to figure these things out.

There have been other publications on increased or tuned climate sensitivities in CMIP6 models (at least Andrews et al., JAMES, 2019, Gettelman et al., GRL, 2019; Mauritsen et al., JAMES, 2019; Zelinka et al., GRL, 2020, but there may be more I'm not aware of). I find it surprising that the authors ignore all of these publications. Their own work needs to be put into the context of these earlier studies.

To estimate the climate sensitivities in different model configurations the authors deviate from the common approach of branching a simulation with instantaneously increased $CO_2$ concentration from a control run with a climate in equilibrium. Instead, the authors start control runs with modified configurations at the same time as the runs with increased $CO_2$ and use anomalies of the latter with respect to the former. The sentence where this is described (L134) cites Andrews et al. (2012) which is a bit misleading as this reference is not appropriate for the use of anomalies. My hypothesis is that likely the use of anomalies is unproblematic, but If there is no reference confirming this I think the authors should show that, e.g. by using other common approaches as

slab ocean models or +4K experiments which also don't require long spin-up runs. I'm a bit concerned about the applied method because of the strong initial (over five years or so) adjustment due to the change of configuration and the change of sign in the temperature trend afterwards.

Minor comments: L60: The table doesn't show "basic differences" between the different EC-Earth versions but version numbers (and resolutions) of the subcomponents. L67: If the conclusion is that the treatment of aerosols is key for the increased climate sensitivity I think that this paragraph describing it should be expanded, for the reader to better understand key properties and differences. L89: I don't like the use of "tas" for near surface air temperature. I know it is a CMIP variable name, but it looks odd in a written text, isn't used for the global mean in CMIP, and is inconsistent with other names (Qnet). L100 "most important updates are likely those related to the revised aerosols". I guess "most important" is meant in the sense of ECS. Why is that likely? Other authors have e.g. documented that also tuning of model physics may affect ECS strongly. Can this be excluded a priori. L117 "Since models may present a not perfectly closed energy balance . . ." Is that the case for EC-Earth? L123 "Therefore we divide . . ." This is common practice. L126 Why is a "well-tuned" model a basic assumption of the Gregory method? And how can good tuning be characterized? L134 The new control experiments are no piControl experiments, which are supposed to start after a spin-up. I'd suggest to name them differently. L185 The authors speak of "subsequent tuning to match a realistic preindustrial equilibrium and present-day climate". This sounds like the model's climate sensitivity was tuned? Or is this just a misunderstanding? L193 I don't find it easy to understand why a change in complexity would have the "potential to modify the sensitivity" beyond the fact that any model change has this potential. I would also like to see an explanation for the statement in the following sentence. Would the assumption be that the addition of an indirect effect would lead overall to a larger aerosol forcing and the attempt to compensate for that by tuning the model to a higher climate sensitivity to obtain a better fit to the historical temperature trend? Code and Data availability: I don't know the exact policy of GMD. But usually

these days journals require the availability of primary data, which to my understanding is the model code and the scripts and input files needed to run the model, not only for editors and reviewers. Table 2: The "experiment" column should contain more information. E.g. it would be nice to be able to identify quickly which experiment in table 3 belong to which in table 2.

———————————————————

---

## Author Comment (AC1) · 31 Mar 2020

Author comments to referees #1 and #2

We'd like to thank the two anonymous reviewers for many helpful comments and suggestions that have helped us to improve the manuscript. Both reviewers acknowledge that the climate sensitivity of CMIP6 models is an important topic, and that it is important to understanding the differences to the models from CMIP5. The reviewers agree in their criticism that the submitted manuscript was incomplete and lacked much of the essential analysis. We have tried to address the reviewers' point by substantially extending the analysis and not only look at the ECS but also at changes in clouds and the cloud radiative forcing, and discuss the impact of these changes to explain the change

in the ECS in the CMIP5 and CMIP6 version of the EC-Earth model. The analysis is now also covering regional aspects and not only global means. We hope the extension of the analysis pleases the reviewers and makes the message of the study more clear. Both reviewers also mention the lack of references to recent papers that discuss climate sensitivity in other CMIP6 models. We agree on that point and have added references to other studies where suitable. However, we'd also like to make a point that both reviewers have given examples of missing references and cite papers that have been published several months after our manuscript was submitted (end of September 2019), and we therefore were not aware of these references (e.g. the excellent paper by Zelinka et al.) We would like to emphasize that our submitted manuscript isn't intended to replace the full documentation of the EC-Earth3 model. The details of the changes when going from the CMIP5 to the CMIP6 version of the model, the model tuning, and changes in the model climate will be documented in a model reference paper. Thus, we have decided to not dive deeper into explicit details of individual steps in the model development, but keep the discussion at a general level and focus on the impact of the developments on the ECS instead.

Detailed replies to the comments from reviewer 1:

Major comments 1) The paper could be easily improved by describing the changes to the model code in terms which can be understood by non-aerosol specialists, e.g. somebody who might be interested model weighting and query your paper for understanding why ECEarth model version X has a higher sensitivity that EC-Earth model version Y; which of the two to include in a certain assessment and how much weight to give each model version. Such a person should be able to take more away from your paper than ". . . [the change] is mainly caused by the change in the representation of aerosols and their impact on clouds and radiation" (line 179), isn't it? It would help to flesh out line 65-88 (not necessarily in length but in readability for non-experts). We have reworked the model description (2.1) and describe the differences between ECE2 and ECE3 clearer. As stated above, it's not the intention that this study replaces

a detailed model description, we try to keep the description at a general level.

2) I my eyes, a main conclusion of the paper is "This series of sensitivity experiments suggests that the increase of the ECS from CMIP5 to CMIP6 is mainly caused by the change in the representation of aerosol and their impacts on clouds and radiation. The implementation of MACv2-SP as it is suggested for CMIP6 models without explicit aerosol scheme has fundamentally changed the way how aerosols are prescribed in the model, yet this change has little effect on the ECS as long as cloud effective radius and autoconversion are independent of the aerosol concentration. The ECS increases when the more advanced treatment of the first and second indirect effect is introduced, with the largest contribution coming from the latter" (line 177-184 or so) The paper would profit from putting actual numbers behind these claims, which are now only expressed in Table 1 and 2 (the only actual results of the paper). Simply showing the relative contributions locally, some spatial fields of how the second indirect effect is expressed, and the different contributions (short and long wave, clear and full sky) to Delta Q net (as in Fig. 1), which is standard in climate sensitivity papers, e.g. Andrews et al. 2012 or 2015, or Zelinka et al. 2020 "Causes of higher climate C2 sensitivity in CMIP6 models" (which goes much deeper in the discussion but does not replace papers like this one). However, there should be more depth than "it's likely the second indirect effect". Why? Where? How trustworthy is it (which model versions should I use for which purpose)? Why did you include it into the model? What do we learn from it about climate sensitivity and projections of warming (and precipitation) in the next century? With the men power of six co-authors you should be able to go into a little bit more depth (e.g. compare Gettleman 2019 "High Climate Sensitivity in the Community Earth System Model Version 2 (CESM2)". We have extended the analysis and now include also discuss the differences in ERF and feedback parameter between between ECE2 and ECE3. We also look at the differences in the sensitivity of clouds and the cloud forcing in the two model versions, and show that the largest difference are found over the Northern Hemisphere Atlantic and Pacific Ocean.

3) The paper discusses changes in ECS of 1K and smaller changes between the model versions. There should be uncertainty estimates for all ECS estimate and a discussion in how far differences are even statistically detectable (I'm guessing the difference in line 155 between 4.3 and 4.2K is not even significant (as you also argue, but you should show it)). We have added uncertainty estimates to all numbers in the table with estimates given as standard deviation of the parameter estimates in the linear regression of the Gregory method.

Minor comments line 21: delete "easily" (of course this can't be generalized to any other model (?)) Agree.

Line 27: what's "the climate change context" (?) Replaced with "...widely used metric in climate modeling..."

line 30: . . . for the warming. I guess in the next century? Are there more recent papers on this discussion? No, the warming in this context is not bound to a fixed point in time (e.g. end of the century). The idea is that the equilibrium temperature at any point in time is determined by the cumulative emission up to this point. To our knowledge, Matthews et al were the first to describe this and we therefore think it makes sense to cite their work. Since then many papers have appeared in the literature that look at the relationship between cumulative emissions and temperature change (e.g. Frölicher 2016, Seshadri 2017, Miller and Friedlingstein 2018, Matthews et al 2018), but we consider the discussion of this topic to be outside the scope of our work and therefore don't add more references.

line 34: "was found" and "has been found" sounds as if these models fell from the sky onto your desk. The models were made more sensitivity (by people like you, changing the model), this is active human action and doesn't passively happen. We haven't tuned the ECS in the EC-Earth model and get its value first after having done some of the CMIP6 simulations. So yes, for us it "fell from the sky". The same is probably true for other modeling groups too, otherwise there wouldn't have been so much discussion

about higher ECS in the new climate models about a year ago at the CMIP6 analysis workshop in Barcelona (which among other things resulted in the Carbonbrief guest post: https://www.carbonbrief.org/guest-post-why-results-from-the-next-generation-of-climate-models-matter)

line 40: "An important question is if we understand the reason for this increase" This statement is very sadly showing the level of climate science these days. I hope you are able to understand the reason for this increase and both the user and the wider public want to know it. If you don't understand the reason, why should I trust you, your model, the IPCC assessment using the model, . . . ? This sentence has been removed.

line 43: "Unfortunately the complex nature of the model development process makes it impossible to turn back the development in a system- C3 atic and continuous approach." What about starting systematically in the first place (!?) *You* developed the model and made these decisions only a few years or even months ago. I know this is a hard process, but the necessity for careful documentation has been openly discussed now for several years (e.g. Hourdin et al., 2016, "The art and science of climate model tuning.", but also already Mauritsen et al. 2012 "Tuning the climate of a global model"). You at least have to blame yourself (or colleagues) for following this approach and not take it as passively given. The EC-Earth model is developed by a broad and heterogeneous consortium. Some developments can be easily reverted, others not. Sometimes it's just not possible (or very difficult) to maintain different versions of process descriptions in the same model. This has nothing to do with model tuning.

line 56: This is the last time I'm making that comment: "The EC-Earth global climate model has evolved from . . ." You did evolve the model (actively) and should be able to confidently explain me why things change, isn't it? This sentence was there to document the legacy of the EC-Earth model, not to explain any of its characteristics. Nevertheless we have removed this sentence as it's not necessary for the ECS.

line 71: Explain how (what is in Martin 1994). line 71-73: More explanation and depth

is necessary for a reader not familiar with the model or aerosols in general We have added a short sentence about the Martin et al parameterization of the effective radius. It would be beyond the scope of this study to go into the technical details of this – or other – parameterisations, the interested reader can easily find the formulae in the cited references.

line 75: Again more explanation necessary. How are these plumes prescribed? Do they change in time in the historical but not in the step forcing (the same way as in the control simulation)? A non-aerosol expert would profit here from some plots visualizing the changes in the model. The aerosol concentrations do not change in time, neither in the pre-industrial control nor in the 4xCO2 simulations. We have added a statement that the aerosol concentrations don't change over time (l.89f in the revised manuscript)

line 79: the *direct* aerosol effect? What's the background aerosol mass concentration? The background aerosol concentration is from the off-line run with TM5, this is described in l.85ff

line 81: shortly explain effective radius, auto conversion-efficiency, activation scheme in laymen's terms We don't think it is necessary to explain these basic concepts of coud microphysic and radiative transfer in a paper about climate model sensitivity.

line 91: What's the baseline $CO_2$ concentration? Is it the same in all models in CMIP6? I think that wasn't the case for CMIP5? How did you deal with this in your model development? Yes, the prescribed $CO_2$ concentration is the same in all concentration driven models, this has been the case for CMIP5 and it still is for CMIP6.

line 126: No that's not a basic assumption of the Gregory method. It is very common to detrend models, e.g. Proistosescu and Huybers 2017 or Andrews et al. 2015. (rest of the section is fine though) Agree, the detrending is widely used and we have removed this statement.

line 139: In how far does discarding the first five years change the results? See comments on uncertainty above. The ECS estimates are changing a lot when the first few years are included in the regression, but we don't find any difference if we exclude 5 or 10 years. We have added a sentence about that (l.148)

line 169: Does the lower value refer to the fixed vegetation version? The paragraph has been reworded.

line 181: as long as cloud *droplet(?)* effective radius . . . Added droplet.

line 184: What's the reason for the correlation in the Kiel 2007 paper? Are there any updates of the discussion in the literature? line 183: This is great that the tuning is discussed Kiehl et al argue that tuning change the ECS of a model, and we are not opposing their finding. However, our sensitivity experiments show that the ECS can also change without changing the tuning of the model, and this is the point we like to make here because we heard the argument in the discussion that higher ECS is just the result from the tuning.

line 189: add maps of these quantities or short wave cloud radiative effects or other feed- C4 backs here (?) Figs 2+3 have been added

line 205: delete "strictly speaking" (or explain how "loosely speaking" your results are valid for any other model (of course they are not (?))) This sentence has been changed according to the reviewer's suggestion.

line 310: Express resolution also in approximate km units Done.

References to statements missing throughout the text (e.g. line 62, whole section 2.3, especially line 110, line 113, line 124, and other places) Sorry, but we don't understand what the reviewer is asking here? For example l.110 reads "concentration until it reaches equilibrium. However, the brute force approach to run the model until equilibrium is not very", what statement on that line would require a reference?
* * *
[Figure]

2019.

---

## Author Comment (AC2) · 31 Mar 2020

"On the increased climate sensitivity in the EC-Earth model from CMIP5 to CMIP6" by Klaus Wyser et al. Author comments to referees #1 and #2

We'd like to thank the two anonymous reviewers for many helpful comments and suggestions that have helped us to improve the manuscript. Both reviewers acknowledge that the climate sensitivity of CMIP6 models is an important topic, and that it is important to understanding the differences to the models from CMIP5. The reviewers agree in their criticism that the submitted manuscript was incomplete and lacked much of the essential analysis. We have tried to address the reviewers' point by substantially extending the analysis and not only look at the ECS but also at changes in clouds and the

cloud radiative forcing, and discuss the impact of these changes to explain the change in the ECS in the CMIP5 and CMIP6 version of the EC-Earth model. The analysis is now also covering regional aspects and not only global means. We hope the extension of the analysis pleases the reviewers and makes the message of the study more clear. Both reviewers also mention the lack of references to recent papers that discuss climate sensitivity in other CMIP6 models. We agree on that point and have added references to other studies where suitable. However, we'd also like to make a point that both reviewers have given examples of missing references and cite papers that have been published several months after our manuscript was submitted (end of September 2019), and we therefore were not aware of these references (e.g. the excellent paper by Zelinka et al.) We would like to emphasize that our submitted manuscript isn't intended to replace the full documentation of the EC-Earth3 model. The details of the changes when going from the CMIP5 to the CMIP6 version of the model, the model tuning, and changes in the model climate will be documented in a model reference paper. Thus, we have decided to not dive deeper into explicit details of individual steps in the model development, but keep the discussion at a general level and focus on the impact of the developments on the ECS instead.

Detailed replies to the comments from reviewer 2:

Major comments: The authors present climate sensitivity estimates from a series of simulations including all or subsets of the aerosol related changes between the CMIP5 and CMIP6 versions. Simulation results indicate that as well direct aerosol effects from the new climatology as the inclusion of different indirect aerosol effects contribute to the higher climate sensitivity. However, no attempt is made to understand why and how these effects come about, which I think would be crucial to make this paper useful for readers beyond the notion that differences in climate sensitivities simulated by different EC-Earth version may be related to the treatment of aerosols. There should be an attempt to attribute the changes to different types of climate feedbacks, as well as to identify possible regional patterns and mechanisms that affect the feedbacks. One

could e.g. imagine that the modifications affect ECS rather indirectly, e.g. by modifying the climatological cloud distribution which then reacts differently to an increased GHG concentration. It would be important to figure these things out. We address the concerns with an extended analysis of the differences between the CMIP5 and CMIP6 version of the model by looking at the response of clouds and the cloud forcing, see new Section 3.1.

There have been other publications on increased or tuned climate sensitivities in CMIP6 models (at least Andrews et al., JAMES, 2019, Gettelman et al., GRL, 2019; Mauritsen et al., JAMES, 2019; Zelinka et al., GRL, 2020, but there may be more I'm not aware of). I find it surprising that the authors ignore all of these publications. Their own work needs to be put into the context of these earlier studies. Indeed, there are many publications about changes in the climate sensitivity of CMIP6 models. We made an attempt to update references in the paper and put our results in a wider context.

To estimate the climate sensitivities in different model configurations the authors deviate from the common approach of branching a simulation with instantaneously increased CO2 concentration from a control run with a climate in equilibrium. Instead, the authors start control runs with modified configurations at the same time as the runs with increased CO2 and use anomalies of the latter with respect to the former. The sentence where this is described (L134) cites Andrews et al. (2012) which is a bit misleading as this reference is not appropriate for the use of anomalies. My hypothesis is that likely the use of anomalies is unproblematic, but If there is no reference confirming this I think the authors should show that, e.g. by using other common approaches as slab ocean models or +4K experiments which also don't require long spin-up runs. I'm a bit concerned about the applied method because of the strong initial (over five years or so) adjustment due to the change of configuration and the change of sign in the temperature trend afterwards. We are sorry for having put the wrong reference for the detrending of the results before evaluating the ECS, the correct reference is Andrews at el (2015) and this has corrected. We are aware that we deviate from the

standard protocol by not letting the model run to equilibrium in its different configurations before starting the 4xCO2 experiment. We explain all this in the manuscript and motivate it with the expensive computational resources that would be needed to run the model to equilibrium for each configuration. Please not also that this deviation from the CMIP6 protocol is only used for the sensitivity experiments in Sec 3.4, the piControl and 4xCO2 experiments for CMIP6 have been started from a properly spun-up state.

Minor comments: L60: The table doesn't show "basic differences" between the different EC-Earth versions but version numbers (and resolutions) of the subcomponents. Agree, changed in the text and table caption. L89: I don't like the use of "tas" for near surface air temperature. I know it is a CMIP variable name, but it looks odd in a written text, isn't used for the global mean in CMIP, and is inconsistent with other names (Qnet). We prefer to use CMIP variable names when possible. There are no CMIP names for net radiation or cloud forcing and for these variables we use Qnet and CF.

L100 "most important updates are likely those related to the revised aerosols". I guess "most important" is meant in the sense of ECS. Why is that likely? Other authors have e.g. documented that also tuning of model physics may affect ECS strongly. Can this be excluded a priori. No, tuning may also play a role as suggested by Kiehl (2007), this is mentioned at the end of Sec 3.4. However, here we show that for the ECE3 model we can get back the ECS from ECE2 by switching of the updates in the aerosol-cloud interaction that were not present in ECE2. In all the sensitivity experiments we don't change the model tuning, it only is an effect of having the indirect aerosols effect switched on or off. We also try to be clear that this is only true for the EC-Earth model, we cannot say anything about if this would have the same effect in other models.

L117 "Since models may present a not perfectly closed energy balance : : :" Is that the case for EC-Earth? Many models can experience a small drift, or long-term climate variability in long runs such as piControl. In the CMIP6 version of the EC-Earth model the decadal or even multi-decadal variability is surprisingly strong for reasons yet unknown.

L123 "Therefore we divide : : :" This is common practice. Yes, but still it has to be mentioned here to explain how we get from the 4xCO2 experiment to a value that corresponds to a 2xCO2 experiment.

L126 Why is a "well-tuned" model a basic assumption of the Gregory method? And how can good tuning be characterized? Indeed, "well-tuned" is not necessary for the Gregory method. We have updated the text accordingly.

L134 The new control experiments are no piControl experiments, which are supposed to start after a spin-up. I'd suggest to name them differently. Agree, we now only use piControl where we refer to the proper CMIP6 (and CMIP5) piControl experiment.

L185 The authors speak of "subsequent tuning to match a realistic preindustrial equilibrium and present-day climate". This sounds like the model's climate sensitivity was tuned? Or is this just a misunderstanding? We tuned the model with the goal to get a stable pre-industrial climate and a present-day climate close to observations. We did not tune the climate sensitivity explicitly. However, when the new treatment of aerosols was introduced we had to re-tune the model to again come close to our tuning goals.

L193 I don't find it easy to understand why a change in complexity would have the "potential to modify the sensitivity" beyond the fact that any model change has this potential. I would also like to see an explanation for the statement in the following sentence. Would the assumption be that the addition of an indirect effect would lead overall to a larger aerosol forcing and the attempt to compensate for that by tuning the model to a higher climate sensitivity to obtain a better fit to the historical temperature trend? The reviewer is correct, any model change can change the sensitivity and it's not a priori given that the sensitivity increases in a more complex model. We therefore have removed this paragraph.

Code and Data availability: I don't know the exact policy of GMD. But usually these days journals require the availability of primary data, which to my understanding is the model code and the scripts and input files needed to run the model, not only for editors and

reviewers. Unfortunately we cannot freely distribute the EC-Earth code but are bound by license agreements with the ECMWF (for the IFS code). GMD has accepted this restriction and allows the distribution of the code only to editor and reviewers. Hopefully this restrictive policy of ECMWF will change with the next version of EC-Earth that will be based on OpenIFS and can be distributed more freely.

Table 2: The "experiment" column should contain more information. E.g. it would be nice to be able to identify quickly which experiment in table 3 belong to which in table 2. Tables have been reorganized and split, and should be easier to read now.

―――――――――――――――――――

---

## Author Response (AR2)

**On the increased climate sensitivity in the EC-Earth model from CMIP5 to CMIP6**

Klaus Wyser[1], Twan van Noije[2], Shuting Yang[3], Jost von Hardenberg[4,5], Declan O'Donnell[6], Ralf Döscher[1]

**Replies to comments:**

1. Reviewer #1 commented on the need for careful documentation in response to your sentence that it is difficult to turn back or switch off developments in a systematic way. While I accept that EC-Earth is a consortium model and some developments may be outside of your control, I wonder whether the current experience with changes in ECS from one version of EC-Earth to another may lead to a change in model development strategy? Are there valuable lessons to be learnt from this that might inform the model development process more generally? Given this may be relevant to other models, could I ask you to add a relevant comment? It would also help to address the reviewer's comment further.

*Indeed, it is difficult to roll back individual changes in a large consortium that work in parallel on many different issues. Climate sensitivity wasn't one of our tuning targets and we haven't looked at it during the development of the CMIP6 model version, only after the first DECK simulations have started. However, climate sensitivity is a critical metrics in the whole climate change discussion (Knutti et al (2017) state that climate sensitivity has reached iconic status) and we therefore should strive to pay more attention to it during the development of the next generation of the EC-Earth model. We have added a paragraph at the end of the conclusions summarizing the lessons learned.*

2. In response to Reviewer #1, could I please ask you to provide a very brief explanation for effective radius, auto conversion-efficiency, and activation scheme? While you may not feel it is necessary, if this reviewer felt an explanation was warranted, there may well be other readers of your manuscript that share that view. This could be placed in a short appendix if you thought that it may affect the flow and focus of the relevant section.

*We have added more detailed explanations and equations for the effective radius, the autoconversion efficiency and the aerosol activation scheme.*

3. Reviewer #1 and missing references: While I cannot speak for the reviewer, looking at the lines referred to, I think the reviewer is just asking to add some (or more) justification/support for your statements. Here are some suggestions:
• Line 62: Please add references for CMIP5 and CMIP6 forcings (e.g. Meinhausen et al., Hoesly et al., etc..)
• Line 110: Here I think the reviewer is referring to the challenge of running a coupled model to equilibrium. Can you include references to papers that also refer to this challenge in support of your statements from lines 108-112?

• Line 113: Likewise, is there evidence from other studies that demonstrate that the Gregory approach is a suitable alternative?

40  • Line 124: Can you provide examples of other studies that divide by 2? I know this is common practice but again, it's just to provide additional support for the approach taken.

*We have added references for all the statements and cite other papers that use similar approaches for assessing the ECS. We would like to emphasize that the Gregory method has actually been suggested by the IPCC WG1*

45  *AR4 for evaluating the ECS in climate models, so our methodology is neither exotic nor novel but rather common.*

4. Reviewer #2: Use of CMIP variable names. I have to agree with the reviewer here and ask that you change names like tas to near-surface temperature.

50

*We have switched "tas" to "T2m" in the text and in the figures.*

[revised manuscript text omitted]

**EC-EARTH v2.3**

[Figure]

**EC-Earth3-Veg**

[Figure]

**EC-Earth3-Veg**

[Figure]

[Figure]

$\Delta$cloud fraction

$\Delta$LWP [kg m$^{-2}$]

**EC-EARTH v2.3**

[Figure]

**EC-EARTH v2.3**

[Figure]

**EC-Earth3-Veg**

[Figure]

**EC-Earth3-Veg**

[Figure]

[Figure]

$\Delta$cloud fraction

$\Delta$LWP [kg m$^{-2}$]

[Figure]

445

**Figure 2. Response of cloud fraction (left) and LWP (right) to a quadrupling of CO2 in ECE2 (top) and ECE3 (bottom).**

**EC-EARTH v2.3**

[Figure]

**EC-Earth3-Veg**

[Figure]

[Figure]

$\triangle CF \ [W \ m^{-2}]$

**EC-EARTH v2.3**

[Figure]

**EC-Earth3-Veg**

[Figure]

[Figure]

$\Delta CF\ [W\,m^{-2}]$

450

**EC-EARTH v2.3**

[Figure]

**EC-Earth3-Veg**

[Figure]

[Figure]

$\Delta CF$ [W m$^{-2}$]

**Figure 3. As Fig. 2 but for net cloud forcing.**

| | EC-Earth2 (ECE2) | EC-Earth3 (ECE3) |
|---|---|---|
| Atmosphere model | IFS cy31r1 | IFS cy36r4 |
| Ocean model | NEMO2 (OPA9) | NEMO 3.6 |
| Sea-ice model | LIM2 with 1 sea ice category | LIM3 with 5 sea ice categories |
| Resolution — Atmosphere | T159L62 (125 km) | T255L91 (80 km) |
| Resolution — Ocean | ORCA1L42 (1 deg) | ORCA1L75 (1 deg) |

455   **Table 1: CMIP5 and CMIP6 versions of the EC-Earth model family**

| MIP | Model | ECS | $\lambda$ | ERF |
|---|---|---|---|---|
| CMIP5 | EC-Earth2 | $3.34 \pm 0.05$ | $-1.01 \pm 0.03$ | $3.37 \pm 0.13$ |
| CMIP6 | EC-Earth3-Veg | $4.31 \pm 0.08$ | $-0.79 \pm 0.03$ | $3.41 \pm 0.17$ |

**Table 2: Equilibrium climate sensitivity (ECS in K), net feedback parameter ($\lambda$ in W m$^{-2}$ K$^{-1}$), and effective radiative forcing (ERF in W m$^{-2}$) in the CMIP5 and CMIP6 version of the EC-Earth model**

460

| Model | Length (years) | Resolution | ECS | Remarks |
|---|---|---|---|---|
| EC-Earth2 | 150 | T159L62-ORCA1L42 | $3.34 \pm 0.05$ | Used in CMIP5 |
| EC-Earth3-Veg | 150 | T255L91-ORCA1L75 | $4.31 \pm 0.08$ | Used in CMIP6 |
| | 75 | T255L91-ORCA1L75 | $4.27 \pm 0.15$ | Reduced length |
| | | T159L91-ORCA1L75 | $4.03 \pm 0.12$ | Reduced length + reduced horizontal resolution |
| | | T159L62-ORCA1L75 | $4.11 \pm 0.12$ | Reduced length + reduced horizontal and reduced vertical resolution |

**Table 3: Impact of a reduced simulation length and reduced model resolution on the ECS. The ECS value for EC-Earth2 is shown for comparison.**

465

| Experiment | Aerosol direct radiative effect | First indirect effect | Second indirect effect | ECS (K) | Net CF (W m$^{-2}$) |
|---|---|---|---|---|---|

| Prescribed aerosol concentrations from CMIP5 | As for CMIP5 | As for CMIP5 | As for CMIP5 | 3.25 ± 0.07 | -21.54 ± 0.32 |
|---|---|---|---|---|---|
| Aerosols as in CMIP6 | As for CMIP6 | As for CMIP5 | As for CMIP5 | 3.54 ± 0.12 | -21.31 ± 0.34 |
| | As for CMIP6 | As for CMIP6 | As for CMIP5 | 3.81 ± 0.12 | -21.69 ± 0.26 |
| | As for CMIP6 | As for CMIP6 | As for CMIP6 | 4.28 ± 0.12 | -18.07 ± 0.28 |

**Table 4: Sensitivity of ECS and net CF to different realisations of the aerosol-cloud interaction processes. All experiments were done with the low resolution (T159L62) configuration of EC-Earth3-Veg and stretch over only 75 years.**